# Effect of Alkaline Treatment on Mechanical and Thermal Properties of Miswak (*Salvadora persica*) Fiber-Reinforced Polylactic Acid

**DOI:** 10.3390/polym15092228

**Published:** 2023-05-08

**Authors:** S. Ayu Rafiqah, A. F. Nur Diyana, Khalina Abdan, S. M. Sapuan

**Affiliations:** 1Institute of Tropical Forestry and Forest Products (INTROP), Universiti Putra Malaysia, Serdang 43400, Selangor, Malaysia; 95difaz@gmail.com; 2Department of Agriculture and Biotechnological Engineering, Universiti Putra Malaysia, Serdang 43400, Selangor, Malaysia; 3Department of Mechanical and Manufacturing Engineering, Universiti Putra Malaysia, Serdang 43400, Selangor, Malaysia; sapuan@upm.edu.my

**Keywords:** alkaline treatment, DMA, polylactic acid, *Salvadora persica*, tensile, TGA

## Abstract

This study examines the effects of alkaline treatment on the mechanical and thermal properties of miswak fiber-reinforced polylactic acid. The treatment was performed with three distinct concentrations of sodium hydroxide (NaOH): 1 wt %, 2 wt %, and 3 wt %. The difficulties of interaction between the surface of the fiber and the matrix, which led to this treatment, is caused by miswak fiber’s hydrophilic character, which impedes its ability to bind with hydrophobic polylactic acid. FTIR, tensile, TGA, and DMA measurements were used to characterize the composite samples. A scanning electron microscope (SEM) was used to examine the microstructures of many broken samples. The treatment is not yet especially effective in enhancing interfacial bonding, as seen by the uneven tensile strength data. The effect of the treated fiber surface significantly improves the tensile strength of miswak fiber-reinforced PLA composites. Tensile strength improves by 18.01%, 6.48%, and 14.50%, respectively, for 1 wt %, 2 wt %, and 3 wt %. Only 2 wt %-treated fiber exhibits an increase of 0.7% in tensile modulus. The modulus decreases by 4.15 % at 1 wt % and by 19.7% at 3 wt %, respectively. The TGA curve for alkali-treated fiber composites demonstrates a slight increase in thermal stability when compared to untreated fiber composites at high temperatures. For DMA, the composites with surface treatment have higher storage moduli than the composite with untreated miswak fiber, especially for the PLA reinforced with 2 wt % alkali miswak fiber, proving the effectiveness of the treatment.

## 1. Introduction

The performance and properties of composite materials are determined by the characteristics of the individual components and their compatibility at the interface. Natural fibers possess some drawbacks when blending with polymers, which have high moisture content and poor compatibility. Brittle matrices with significant moisture absorption and swelling tend to develop cracks on the composite surface. To effectively improve interfacial interactions and provide desired characteristics, their surface properties must be modified adequately. Alkali treatment on the fiber surface is one alternative to overcome these issues. Alkali treatment can also bring different functional groups to the surface of natural fibers, and these functional groups can create strong covalent connections with the matrix to create a strong fiber/matrix contact. When just moderate mechanical properties are needed, natural fiber composites are a great alternative [1,2]. The use of environmentally friendly materials in technology is growing in popularity. Many research studies on cellulose fibers have recently been conducted [3,4,5,6,7,8]. However, the use of single biodegradable materials such as polylactic acid incurs high cost compared with conventional polymer. Prices of PLA (2.2 $/kg) are found to be higher than those of fossil-based plastics (0.95–1.7 $/kg) and around two times more expensive than PP (1.2 $/kg). Therefore, compounding the biodegradable polymer with natural fiber is an alternative to control the cost of production. This research and these developments continue as economies take accountability on the cost of raw materials and also the processing [9]. Each natural fiber is a composite consisting of a flexible lignin and hemicellulose matrix and a firm cellulose microfibril reinforcement. In addition, the intensity of adhesion between the fibers and the matrix is the most crucial factor in achieving effective fiber reinforcement in composites. The adhesion level of these materials is determined by their structure and polarity. Since many fiber components contain hydroxyl and other polar groups, significant moisture absorption occurs, resulting in poor wettability, limited interfacial contact between fibers, and more hydrophobic matrices. Once the fibers in concern are utilized as fillers in polymer composites, this is undesirable. Regarding natural fibers, a variety of fiber treatments are utilized to modify not just the interphase, but also the fibers’ morphology. The significant advantage of the alkaline treatment is the modification or destruction of the hydrogen bonds that hold the whole system together, resulting in a rougher surface. The alkali treatment of fiber has been the focus of extensive research. In order to properly interlock with the matrix, chemicals may introduce new molecules or activate hydroxyl groups. Developing a comprehensive theory for the mechanism of chemical bonding in composites is a challenging task. Many chemical coupling agents are molecules with multiple functions. The hydroxyl groups of cellulose are interacted with first, followed by the matrix’s functional groups [10,11,12]. Among the various surface treatments used, the alkaline treatment is the most economical. To remove impurities like wax, pectin, hemicellulose, and mineral salts, a purifying treatment was used [13]. The fiber’s texture and, especially, its accessibility within aqueous environments are modified by the treatment. When using natural fibers, however, the mechanical characteristics of a single fiber have a negligible effect on the composite’s overall mechanical performance. The fiber characteristics of composites have an effect on their mechanical qualities. Fibers’ length, diameter, and shape may all alter due to shear stress [14]. Bartos et al.’s [15] study discovered the alkaline treatment of bagasse fibers, which have been demonstrated to improve creep behavior by binding the fibers together. In this research, a polylactic acid composite is employed with miswak fiber (*Salvadora persica*) to strengthen the polymer matrix. Native to the hot, dry environments of Africa, the Middle East, and the Arabian Peninsula, miswak is a tiny evergreen shrub or tree. The natives put it to pharmaceutical use due to its high concentration of health-enhancing active chemicals. While this fiber’s precise properties have yet to be determined, they seem to be similar to those of other naturally occurring fibers. People can eat the fruits of the *S. persica* plant, and they have a tangy taste. Toothbrushes may be made from anything from twigs to roots. The salt and resin content of *S. persica* is supposed to have a cleansing effect, and extracts from the plant are used in a variety of European dental pastes [16]. There are about 18 components in miswak that work together to protect teeth and promote oral health. Salvadorine and trimethylamine are two examples of alkaloids with antibacterial effects. Oral health depends on nutrients such as vitamin C, sulfur, calcium, chloride, and fluoride. Essential oils are odorless, taste neutral, and stimulate saliva production while decreasing flatulence. To stop tooth decay, resins cover the enamel in a protective layer. Tannins are a natural astringent that include trace amounts of saponins, flavonoids, and sterols. They also increase premolar saliva production [17,18,19]. The aim of this work was to explore the possibility of improving the effect of alkaline treatment of miswak fiber on the FTIR, tensile, TGA, and DMA characteristics of reinforced polylactic acid composites.

## 2. Experimental Section

### 2.1. Materials

This miswak stick, supplied by Al-Khair Natural Products was bought from a vendor in Selangor, Malaysia. Figure 1 shows the Miswak fibre plant or known as *Salvadora Persica* plant. The PLA pallets were provided by Polycomposite Sdn. Bhd located in Perlis, Malaysia. Evergreen Engineering & Resources’ from Selangor, Malaysia sodium hydroxide was utilized as a chemical reagent. Table 1 and Table 2 shows properties of PLA and miswak fiber.

### 2.2. Alkaline Treatment of Miswak Fiber

Before the fibers were utilized for purification, they were milled and sieved to a size range of 0.25 to 0.60 mm. After an hour of soaking in 1 wt %, 2 wt %, and 3 wt % sodium hydroxyl (NaOH) solutions at room temperature, the fibers were rinsed with distilled water. Before being dried in an oven for 24 h at 60 °C, the fiber was air-dried for 48 h.

### 2.3. Preparation of Composite Board

The PLA pallets were initially desiccated in a furnace (ESCO ISOTHERM model, Selangor, Malaysia) range temperature 0–150 °C) at 80 °C to prevent excessive hydrolysis, which could compromise the polymer’s physical properties. Then, a counterrotating internal mixer (BRABENDER) from Singapore was used to combine PLA with miswak fiber at 160 °C and 60 RPM for 15 min, or until a homogeneous compound was obtained. Using a pulverization machine, the compound was reduced to small fragments prior to being compressed. The composite was then compressed into a testing board by being molded under a hot press (TECHNOPRESS) for 4 minutes at 160 °C, followed by 3 min of cooling. The specimens were then shaped in accordance with the desired characterization tests. All samples prepared contained 70% PLA and 30% MF. Meanwhile, MF was treated with 0%, 1%, 2%, and 3% alkali treatment.

### 2.4. Characterization

#### 2.4.1. Fourier-Transform Infrared Spectroscopy (FTIR)

FTIR measurements were taken in the wavenumber range of 500–4000 cm^−1^ with 4 cm^−1^ resolutions, and 16 scans were taken using a Perkin Elmer 1600 Infrared spectrometer from Selangor, Malaysia. Nicolet software was used to monitor the locations of major transmittance peaks at various wavenumbers.

#### 2.4.2. Tensile Testing

The composite was tested with tensile testing using a 5 kN Bluehill INSTRON Universal Testing Machine (Selangor, Malaysia). The test was carried out according to ASTM standard D-638. With dimensions of 150 × 23 × 2 mm in length, width, and thickness, the specimen was molded into a dog-bone form using a MT1130 die-cutter (MinEuro, Turin, Italy) equipped with a metallic dog-bone-shaped cutting edge. The sample was gripped using the metallic dog-bone shape before being pushed through the cutter.

The crosshead speed was set at 2.0 mm/min, and the composites were held at a gauge length of 30 mm. All specimens were stored in the conditioning room, and the test was conducted at 22 °C with a relative humidity (RH) of 55%. For each test condition, seven specimens were examined.

#### 2.4.3. Scanning Electron Microscopy (SEM)

The samples’ morphology was examined using a Hitachi S-3400N (Selangor, Malaysia) scanning electron microscope (SEM) with an energy dispersive X-ray (EDX) and an acceleration voltage of 15 kV. To prevent the charging effect, the samples were gold-sputtered prior to observation.

#### 2.4.4. Thermogravimetric Analysis (TGA)

A TA Instruments Q500 thermogravimetric analyzer (TGA) (Selangor, Malaysia) in accordance with ASTM E1131-08 was used to determine the samples’ thermal stability. In a nitrogen gas atmosphere, around 6 mg of sample was heated from 30 to 600 °C at a rate of 20 °C min^−1^. The thermal stability of the samples could be evaluated with the use of the TGA.

#### 2.4.5. Dynamic Mechanical Analysis (DMA)

Using a dynamic mechanical analyzer TA Instruments Model Q800 (Selangor, Malaysia), the dynamic mechanical characteristics of a treated miswak/PLA sheet were studied. The samples measured 46 mm in length, 10 mm in width, and 3 mm in thickness. Tensile tests on samples were performed. The three-point bending test mode was at a constant 1.0 Hz and the temperature was changed from 20 °C to 120 °C in 10 °C/min increments.

## 3. Results and Discussion

### 3.1. Fourier-Transform Infrared Spectroscopy (FTIR)

FTIR spectroscopy revealed the structural properties of untreated and treated MF/PLA composites. The treated and untreated miswak fiber composites’ FTIR spectra are shown in Figure 2, and both exhibit a variety of modifications. The sample’s primary absorbance was seen in low-wavenumber areas, particularly between 500 and 1800 cm^−1^.

The lower peaks indicated that some hemicellulose was removed during the NaOH chemical treatment. This study revealed that lignin and hemicellulose are partially re-moved during the NaOH treatment [20]. In a prior investigation, a similar finding was made [21]. The spectra of three distinct concentrations of alkali treatment of fiber in the matrix are practically identical, as indicated in Figure 2. The composites with 1%, 2%, and 3% alkali treatment showed no obvious peak shifting, indicating that alkaline treatment had no effect on the composite functional groups. In the wavenumber band, most of the peaks are similar.

The typical peaks of PLA, on the other hand, were found between 2800 and 3000 cm^−1^. The peaks at 2994 cm^−1^ correspond to the carboxylic acid O-H stretching and those at 2944 cm^−1^ correspond to the alkane C-H stretching. A broad band spanning from 3000 to 3700 cm^−1^ for untreated MF demonstrated the existence of hydroxyl groups (−OH) in the cellulose structure of fibers that can be clearly identified in any natural fiber-reinforced composite, whereas a band spanning from 2800 to 3000 cm^−1^ confirmed the presence of the methylene group C-H bonds [22,23].

The spectrum of the untreated MF/PLA composite has the main peaks located at almost the same wavenumbers at 1746 cm^−1^, 1451 cm^−1^, 1381 cm^−1^, 1359 cm^−1^, 1266 cm^−1^, 1180 cm^−1^, 1127 cm^−1^, 1042 cm^−1^, and 755 cm^−1^. Meanwhile, the spectra of the treated MF/PLA composites have a similar range of wavenumbers located at 1747 cm^−1^, 1452 cm^−1^, 1381 cm^−1^, 1359 cm^−1^, 1266 cm^−1^, 1180 cm^−1^, 1127 cm^−1^, 1043 cm^−1^, and 755 cm^−1^. The treated fiber composites show weak intensity between 2800 and 3000 cm^−1^, indicating that they are less hydrophilic with removed hemicellulose and lignin. Alkaline treatment changes the physicochemical properties of MF. This phenomenon is connected to the mercerization of natural fibers. During mercerization, the waxy epidermal tissue, adhesive pectin, and hemicellulose that tie fiber bundles together as well as pectin and hemicellulose-rich core sheaths are all eliminated [24,25].

Similar to the untreated MF/PLA composite, no notable peaks or evident shifts were found in the neat PLA spectra, as the interaction between the MF fiber and the PLA matrix is primarily physical in this investigation, although the fiber was treated with NaOH. However, many minor peaks were recorded at wavenumbers of 1747 cm^−1^, 1452 cm^−1^, 1381 cm^−1^, 1359 cm^−1^, 1266 cm^−1^, 1180 cm^−1^, 1127 cm^−1^, 1042 cm^−1^, and 755 cm^−1^. The C-O stretching of ester or tertiary alcohol groups of hemicelluloses can be assigned to both. The availability of carbonyl groups is evident in the presence of hemicellulose in fibers, as evidenced by a strong peak at 1747 cm^−1^ attributable to the C=O stretching of carbonyl groups in both untreated and treated MF composites. The peak at 1452 cm^−1^ reveals the presence of a lignin structure in untreated MF composites; however, these peaks appeared weak in all treated MF composites following alkali treatment, indicating that NaOH partially dissolved low molecular components from fibers. Hemicellulose and lignin were partially eliminated in all treated MF/PLA composites [26,27,28].

The effect of alkali treatment on lowering the moisture absorption of natural fiber polymer composites has been documented by a number of studies [29,30]. During surface treatment, the surface of the fibers is cleaned to remove contaminants, lignin, oils, and waxy substances to increase the fiber surface roughness and to reduce moisture absorption by removing the OH group from the fibers [29,31].

Data from a previous study showed that after being treated with alkali, the carbonyl (C=O) stretching vibration peaks at 1735 cm^−1^ in both the alkali-treated bamboo fiber/PLA composite and the dopamine-treated bamboo fiber/PLA composite were eliminated. Research indicates that in the region of the carbonyl absorption peak, some hemicellulose groups dissolve when exposed to an alkaline solution [32]. An increase in infrared transmittance at 1180 cm^−1^ following alkali treatment is clearly indicative of an increase in aromatic ether (C-O) and a reduction in hemicellulose and lignin. In conclusion, miswak fiber treated with the appropriate alkali concentration may improve the composites’ interfacial properties by dissolving impurities, wax, hemicellulose, and lignin.

### 3.2. Tensile Properties

Tensile characteristics of miswak fiber-reinforced PLA composites are shown in Figure 3 and Figure 4. The impact of fiber-surface modification on the tensile strength of treated miswak fiber-reinforced PLA composites is noticeable.

PLA obviously has the greatest tensile strength, measuring 46.3 MPa. Nonetheless, it has been shown that NaOH treatment enhanced the composite’s tensile properties. Alkali treatment improved the tensile strength of the composites by 18.01%, 6.48%, and 14.50% compared to the untreated miswak fiber-reinforced PLA. The 1% alkali-treated composite, which has the highest value of all the treated composites at 28.4 MPa, demonstrates better stress transfer between the fiber and the matrix. It is possible that the NaOH treatment roughened the fiber surfaces, increasing interfacial adhesion, promoting fiber-matrix interlocking, and eventually optimizing the composites’ mechanical performance [13].

In terms of stiffness, the tensile moduli of the miswak fiber-reinforced PLA composites for alkali fiber treated at 1 wt %, 2 wt %, and 3 wt % concentrations are 2770 MPa, 2910 MPa, and 2320 MPa as compared to about 2890 MPa for the untreated miswak fiber-reinforced PLA composites. Our findings indicate that the tensile moduli of fiber composites treated with concentrations of 1 wt % and 3 wt % decreased. Nevertheless, the modulus result for the 2 wt % alkali-treated fiber composite is 2910 MPa, demonstrating that this treatment makes the fiber in the composite stiffer than previous treatments. These results show how significantly fiber surface treatment affects the mechanical properties of the PLA composite reinforced with miswak fiber. Karsli et al. [33] reported that when alkali treatment is applied to fiber surfaces, the impurities at the fiber are removed and the diameter of the fiber is reduced. Thus, as a result of the alkali-treated miswak fibers’ increased stiffness, the fiber stiffness rises, and the moduli of the composites improve.

The elongation-to-failure of PLA composites reinforced with miswak fibers is shown in Figure 4 for both the untreated and treated samples. Alkali-treated miswak fiber composites often have much higher strengths and moduli than untreated fiber composites. The elongation-to-failure of the 1 wt % alkali-treated miswak fiber composite was higher than that of the untreated miswak fiber composite. This effect was caused by the miswak fiber’s alkali treatment, which removed lignin, pectin, and other contaminants from its structure. As a consequence, the presence of cellulose could be seen, enhancing the pliability and flexibility of the composite materials. Nevertheless, fibers treated with 2 wt % and 3 wt % exhibit a declining trend. That is probably because the amorphous components, such hemicellulose and lignin, were only partially removed at a short treatment period and low alkali concentration, leaving behind the remainder. Moreover, the alkali treatment at this low alkali concentration and short treatment interval was insufficient to properly develop the surface roughness that was critical to the adhesion strength [34].

### 3.3. Scanning Electron Microscopy (SEM)

Figure 5 and Figure 6 demonstrate the SEM micrographs of treated and untreated bio-composite samples following tensile testing.

Figure 5a,b depict the cracked surface of the miswak fiber as well as the interaction between the miswak fiber and the PLA matrix. Figure 6a shows the smooth and rough surface of the 1 wt. % alkali-treated miswak fiber composites. The treated miswak fiber-reinforced PLA composites with rougher surfaces experienced physical microstructural changes, as shown in Figure 5b. The treated miswak fiber and PLA matrix adhere to one another more effectively by removing impurities from the fibers during the alkali treatment procedure.

Figure 6a shows a scanning electron micrograph of a sample of the untreated fiber-reinforced PLA composites. The figure illustrates the presence of surface voids as well as fiber rupture. Matrix/fiber adhesion problems may be seen in the form of fiber pull-out.

As can be seen in Figure 6b for the treated miswak fiber-reinforced PLA composites, a good wettability between the PLA matrix and the treated miswak fibers enhances adhesion at the interface of the PLA and miswak fibers. According to the findings, the mechanical characteristics of the bio-composites were improved by increasing the roughness of the fiber surface, which improved the adhesion between the PLA and the miswak fiber interface. These results indicate that the inclusion of alkaline treatment enhances the adhesion between PLA and miswak fiber. By increasing the bio-composite fiber surfaces’ physical roughness, the interface results in improved mechanical properties [35].

### 3.4. Thermogravimetric Analysis (TGA)

Figure 7a displays the improved thermal stability that results from alkali treatment of pure PLA and miswak fiber-reinforced PLA composites, respectively. Thermal decomposition of the composites occurs between 25 °C and 600 °C, with thermal degradation occurring in three phases. Below 100 °C, water molecules evaporate or dehydrate, causing the first weight loss in both untreated and treated MF/PLA bio-composites [36]. Degradation of cellulosic and non-cellulosic materials occurs in a second stage between 250 and 350 °C. The final decomposition of materials occurs in the third stage between 450 and 550 °C. Untreated MF/PLA bio-composites with 1 wt %, 2 wt %, and 3 wt % MF/PLA show 92.4%, 93.5%, and 92.1% thermal degradability at 397 °C, 384 °C, and 389 °C, respectively, with residue weights of approximately 7.3%, 5.8%, and 7.6%. The fiber composite treated with 2% alkali has the lowest thermal stability.

The weight loss for the PLA is 98.4%, whereas the weight of the residue is only approximately 0.8%. The weight loss for an untreated fiber composite is 85.8%, while the weight of the residue is around 14.2%. Alkali treatment therefore enhanced the thermal stability of natural fiber. This is where the hemicellulose, lignin, and glycosidic connections of cellulose in natural fibers thermally degrade [36]. There may have been some readily hydrolyzed components removed from the treated MF/PLA bio-composites that broke down at temperatures lower than the major constituents, since the weight loss reduced with increasing temperature [29]. Figure 7b displays the results of a thermal study of the derivative weight loss curves of pure PLA, untreated PLA, and treated miswak fiber-reinforced PLA composites.

The weight loss (DTG) vs. temperature charts in Figure 7b also show the rate of decomposition of the different composites. Even though the untreated miswak fiber composite degrades thermally at 317 °C, neat PLA degrades thermally in one step at 403 °C. The degradation temperature of composites using untreated miswak fiber is lower than that of pure PLA. The thermal degradation temperature was influenced by the addition of miswak fiber to the PLA matrix. This could be due to the fact that composites are processed by the mixer’s screw, and the molecular weight of PLA reduces. The major peaks, which represent the fastest rates of weight loss, are clearly visible. The breakdown peaks of the treated MF/PLA bio-composites shift towards higher temperatures, showing a derivative weight loss rate of 5% per minute, whereas the untreated MF/PLA bio-composites exhibit a low rate of weight loss. The rate of degradation reveals the breakdown of cellulose and hemicellulose in bio-composites as well as the existence of voids or fiber loosening [37]. As compared to composites without surface treatment, the composite with surface treatment has greater thermal stability. The interfacial adhesion could be enhanced, and the temperatures required for thermal degradation can increase as a consequence of the chemical interaction between the PLA matrix and miswak fiber. Moreover, the composites with alkali-treated fiber have a stronger thermal degradation temperature than composites with untreated fiber. The results indicate that the interface bond between miswak fibers after being alkali-treated is more stable than the interface bond between miswak fibers that have not been treated.

### 3.5. Dynamic Mechanical Analysis (DMA)

Dynamic Mechanical Analysis (DMA) is a technique for obtaining information about a material’s physical structure by utilizing viscoelastic mechanical properties. This test can be used to determine the material’s response to a sinusoidal force during a temperature or frequency sweep. DMA can be used to identify the composite’s mechanical properties (mechanical modulus or stiffness and damping) and critical thermal transitions in the adhesive, such as the glass transition temperature. DMA and DSC testing can be coupled to develop a rapid and easy test technique for comparing a proposed new raw material [38,39]. Table 3 tabulates the values for the storage modulus, loss modulus, and tan δ of untreated and alkali-treated MF/PLA composites.

Figure 8a,b and Figure 9 demonstrate how the dynamic storage modulus, loss modulus, and tan delta of pure PLA, untreated MF/PLA, and treated MF/PLA composites change as a temperature increases or decreases. For untreated MF, the E’ value is lowest, which is 1.5 GPa. For 1%-, 2%-, and 3%-treated MF, the value of the storage modulus increases by 20%, 60%, and 13.3%.

The addition of treated MF to bio-composites improved the E’ value considerably, as the temperature increased for all bio-composites. This suggests that the fiber strengthens the PLA’s ability to withstand mechanical constraints with recoverable viscoelastic bend. With the inclusion of fiber, the storage modulus increased significantly in the glassy zone, whereas there was little change in the rubbery region. In the glassy zone, the components are highly immobile and closely packed, resulting in a high storage modulus value below the Tg. The components, on the other hand, become more mobile and lose their tight packing in the rubbery area, with no significant change in E’. The DMA findings reveal how heating the composite sample might affect their stiffness [17]. The storage modulus has a strong relationship to a material’s load-bearing capacity. Figure 8a demonstrates that the storage moduli of the alkali-treated MF/PLA composites are higher than that of the PLA matrix. When compared to the clean PLA specimen, the higher E’ for all fiber-reinforced bio-composites represents their comparatively high mechanical properties. The static mechanical behavior, which varies depending on the fiber ratio and characteristics, is consistent with these findings.

This may be because the fibers’ higher stiffness of the reinforcement allows for more stress transfer from the matrix to the fiber [14]. In Figure 8a, it is evident that the composites with surface treatment have greater storage moduli than the composites with untreated miswak fiber, particularly for the PLA reinforced with 2% alkali-treated miswak fiber. This shows that the alkali treatment improves the adhesion between the PLA matrix and the miswak fiber compared to the untreated miswak fiber. Since the composite that underwent alkali treatment has the highest storage modulus, alkali treatment appears to be an effective method for improving the adhesion between the PLA matrix and miswak fiber. All samples’ storage moduli significantly decrease as the temperature rises, with the areas between 55 °C and 70 °C observing the greatest drops. However, compared to PLA, the softening temperatures are greater with surface treatments. It could be the consequence of an interaction that causes a regular reinforcing effect and a reduction in chain mobility [20].

Figure 8b displays the loss modulus of composites made of plain PLA, untreated MF/PLA, and treated MF/PLA. The temperature (Tg) of the system is where the greatest amount of heat is dissipated. The glass transition temperature (Tg) of the composites increases after NaOH treatment. The Tg of the plain PLA is 66 °C and decreases to 63 °C when MF is added. However, the Tg value increases by 9.5%, 7.9%, and 6.3% when treated MF is added. The surface treatment partially dissolved lignin and hemicellulose contained in the fiber, thus increasing the amount of exposed cellulose, and owing to this, the crystallinity of MF was modified with alkali treatment. With increasing crystallinity, the modulus of PLA also increases. The correlation between an increase in Tg and a reduction in the mobility of the matrix chains is indicative of increased interfacial adhesion between the fiber and the matrix [36].

The ratio of elasticity (storage modulus) to viscosity (loss modulus) properties (tan = E’/E”) is the damping factor (tan δ) of a material. As a result, a high value suggests a significant degree of energy dissipation and non-elastic deformation. The damping factors of untreated and treated MF/PLA bio-composites are shown in Figure 9. The PLA has the greatest peak value, indicating that it has a higher degree of molecular mobility. Within the range of 55–70 °C, narrow and abrupt peaks of tan δ curves are due the addition of fibers, which greatly reduce the PLA matrix’s damping factor; untreated MF/PLA bio-composites have a lower tan δ peak height than treated MF/PLA bio-composites. It was obvious that the fiber reinforcement stifled the PLA chains’ mobility. Tan δ remains zero at relatively low temperatures (<50 °C), which indicates that the internal damping is zero. The size of MF is too macro to affect their mobility as compared to the molecular segments. The MF showed a very limited effect on the Tg of the polymer matrix. The slightly decreased Tg of the PLA/MF composites is due to the alkaline treatment that improves the molecular mobility of the PLA matrix. When the temperature rises, tan δ increases sharply due to a significant increase in viscous dissipation when materials transfer from the glassy state to the high-elastic state. When comparing the temperature corresponding to the peak value of tan δ, it was found that the Tg of the MF/PLA composites shifted slightly to lower temperatures.

From about 50 °C to 65 °C, the E’ curves showed a rapidly declining trend, indicating a glass/rubbery state transition. Savas (2019) [40] found a similar abrupt declining trend in the modulus curves of miswak powder-reinforced polypropylene composites. Furthermore, the decrease of E’ in fiber-filled composites owing to temperature increase was more significant for composites with higher miswak content, since at higher miswak content, the fiber inclusion could become aggregated, thus lowering the modulus. According to Jamshaid et al., 2023 [41], the insertion of fibers slows down the energy dissipation process by lowering the intensity of tan δ peaks. This is consistent with PLA and its bio-composites’ storage and loss modulus. In a polymeric structure, the material’s damping qualities maintain the right balance between the elastic phase and viscous phase. A strong contact will withstand more strain and lose less energy. In contrast to a material with a highly connected interface, composite materials with inadequate interfacial bonding would often dissipate more energy and have a large damping peak amplitude. Comparative results have also been published, including PLA reinforcement with date palms (Abu-Jdayil et al., 2021) [42], durian skin fiber (Manshor et al., 2014) [43], and miswak powder (Savaş, 2019) [40]. The tan δ plot also revealed that when the temperature rises, damping rises to its maximum in the glassy zone before falling in the rubbery region.

## 4. Conclusions

Miswak fiber-reinforced composites’ mechanical and thermal characteristics were affected by alkaline treatment. The functional groups of the miswak fiber are only affected by the fiber treatment method. With PLA functioning as a control, a comparison between an untreated fiber composite and composites with varied amounts of treated fiber is carried out. The sample’s primary absorbance was found to be between 500 and 1800 cm^−1^ in low-wavenumber zones. Tensile strength, Young’s modulus, and elongation-at-break were used to evaluate the mechanical characteristics of the composite. The findings demonstrated that the untreated fiber-reinforced PLA composite had a weaker tensile strength than pure PLA. Yet, the tensile strength of alkali improved after fiber treatment. The morphology of the composite was examined using an electron scanning microscope. The tensile fracture specimens of the PLA matrix, untreated MF composite, and treated MF composite were examined using scanning electron microscopy. As revealed by SEM, tensile failure was due to non-homogeneous treated fiber, poor wettability, and feeble interfacial adhesion between the fiber and the matrix. Utilizing thermo-gravimetric analysis, the thermal stability of the composite was monitored. TGA curves for untreated and treated MF/PLA composites reveal a single step of thermal degradation. The results indicate that treated MF/PLA composites are superior to untreated MF/PLA composites in terms of weight reduction and temperature stabilization. The DTG peaks represent the degradation temperatures of untreated and treated MF/PLA composites, respectively. At temperatures between 250 °C and 450 °C, individual peaks were observed. Using the storage modulus, loss modulus, and damping, the thermo-mechanical properties were measured. The storage modulus (E’), which dropped as the temperature increased for both treated and untreated composites, was dramatically increased by the inclusion of miswak fiber. Each loss modulus (E”) curve indicated a minor peak with magnitude and temperatures similar to the PLA peak, indicating that the loss modulus of MF in PLA composites varies as a function of temperature (°C). When the damping factors of untreated and treated MF/PLA composites were compared, the PLA showed the greatest peak value, suggesting a greater degree of molecular mobility.

## Figures and Tables

**Figure 1 polymers-15-02228-f001:**
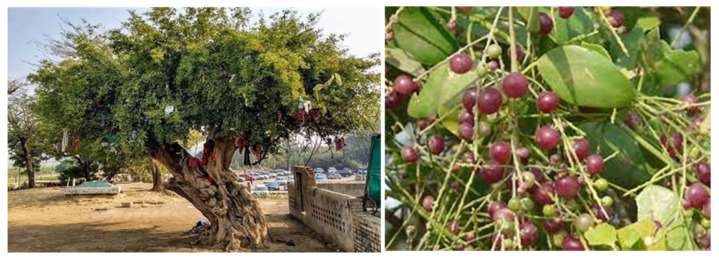
*Salvadora Persica* plant (https://forestrypedia.com/salvadora-persica/, accessed on 8 April 2023).

**Figure 2 polymers-15-02228-f002:**
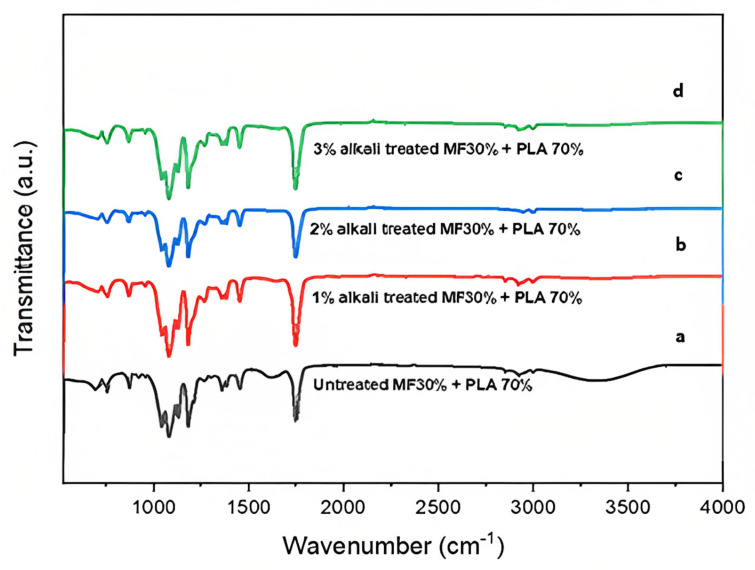
FTIR spectra showing (**a**) untreated; (**b**) 1% alkali-treated; (**c**) 2% alkali-treated; (**d**) 3% alkali-treated miswak fiber-reinforced PLA composites.

**Figure 3 polymers-15-02228-f003:**
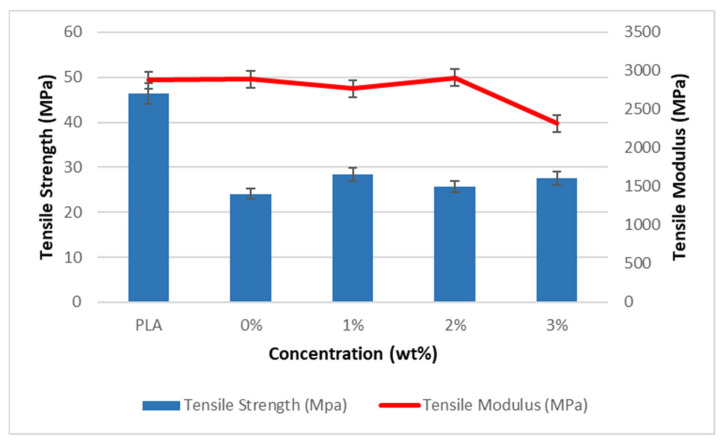
Tensile strength and moduli of miswak fiber-reinforced PLA composites.

**Figure 4 polymers-15-02228-f004:**
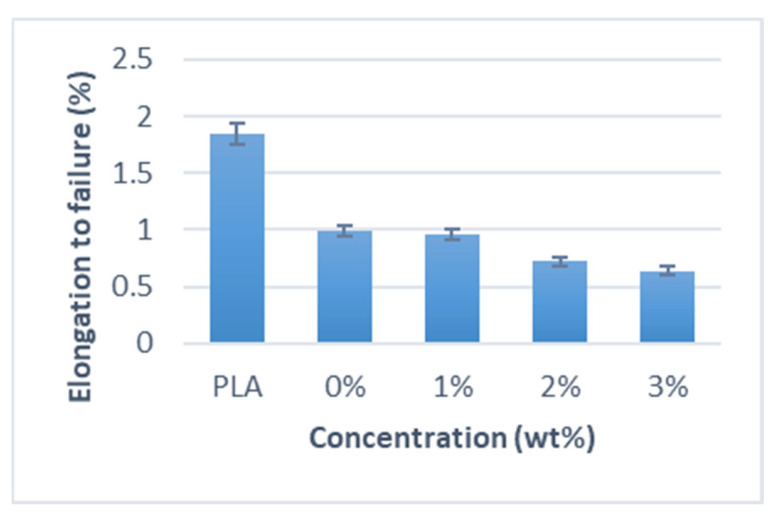
Elongation to failure of miswak fiber-reinforced PLA composites.

**Figure 5 polymers-15-02228-f005:**
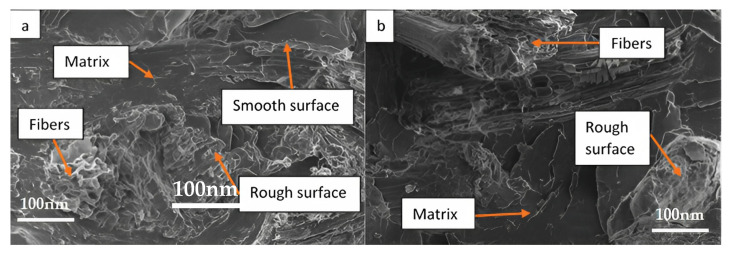
SEM micrographs of (**a**) 1% alkali-treated and (**b**) 2% alkali-treated miswak fiber-reinforced PLA composites following tensile test.

**Figure 6 polymers-15-02228-f006:**
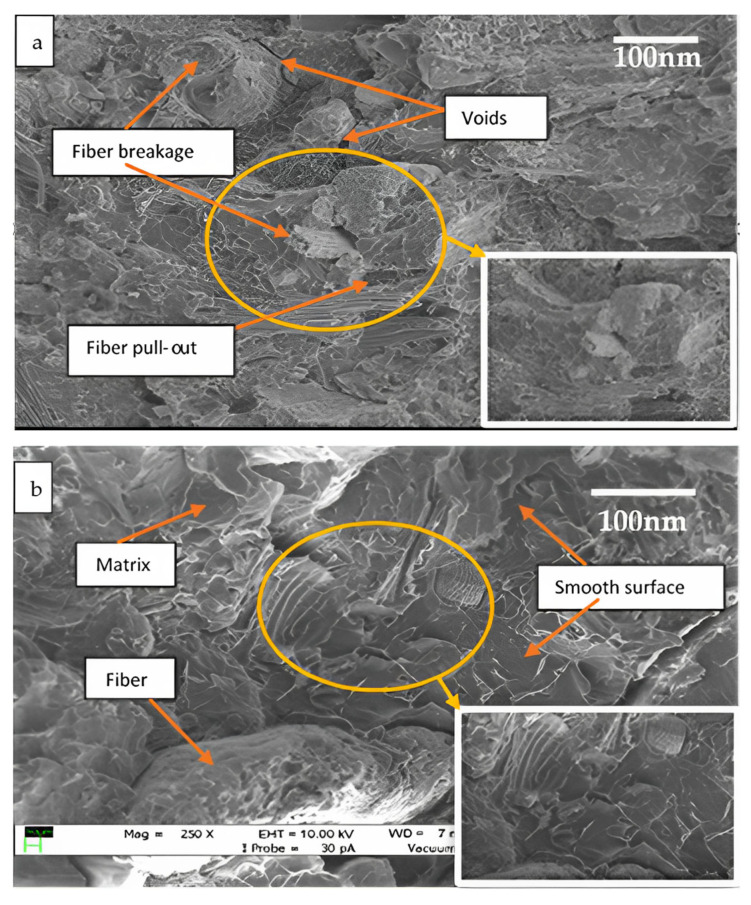
(**a**) SEM micrograph of untreated miswak fiber-reinforced PLA composites following tensile test (**b**) SEM micrograph of 3% alkali treated miswak fiber-reinforced PLA composite following tensile test.

**Figure 7 polymers-15-02228-f007:**
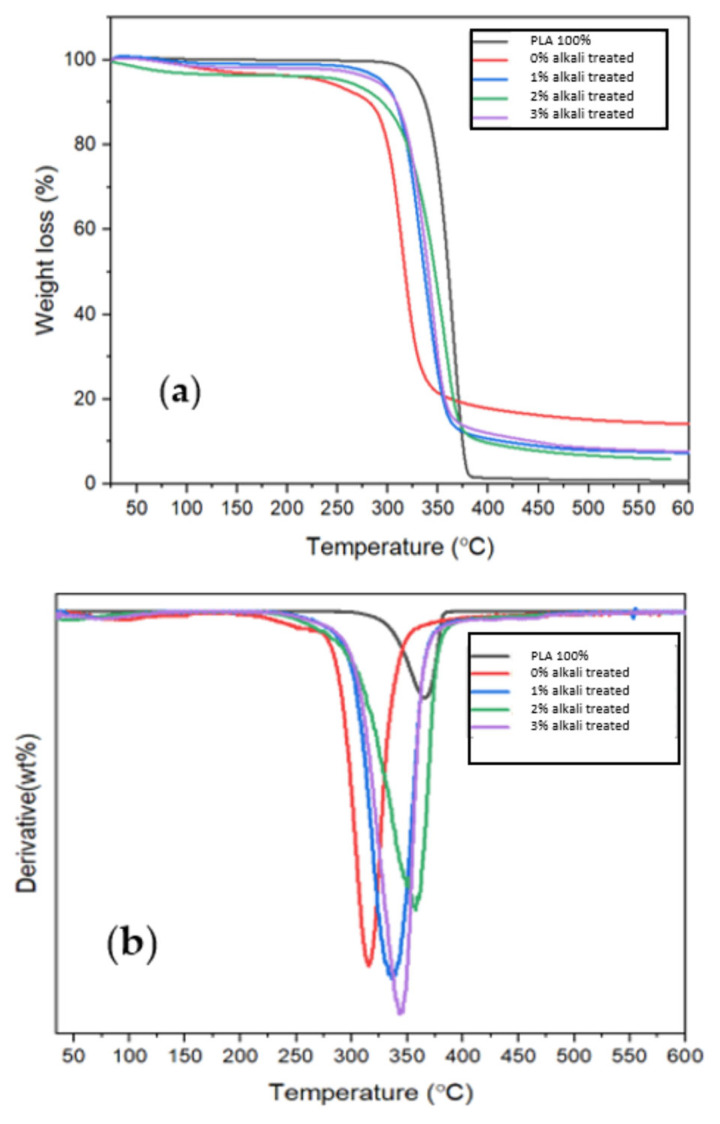
(**a**) TGA curve of pure PLA and MF/PLA composites; (**b**) DTG curve of pure PLA and MF/PLA composites.

**Figure 8 polymers-15-02228-f008:**
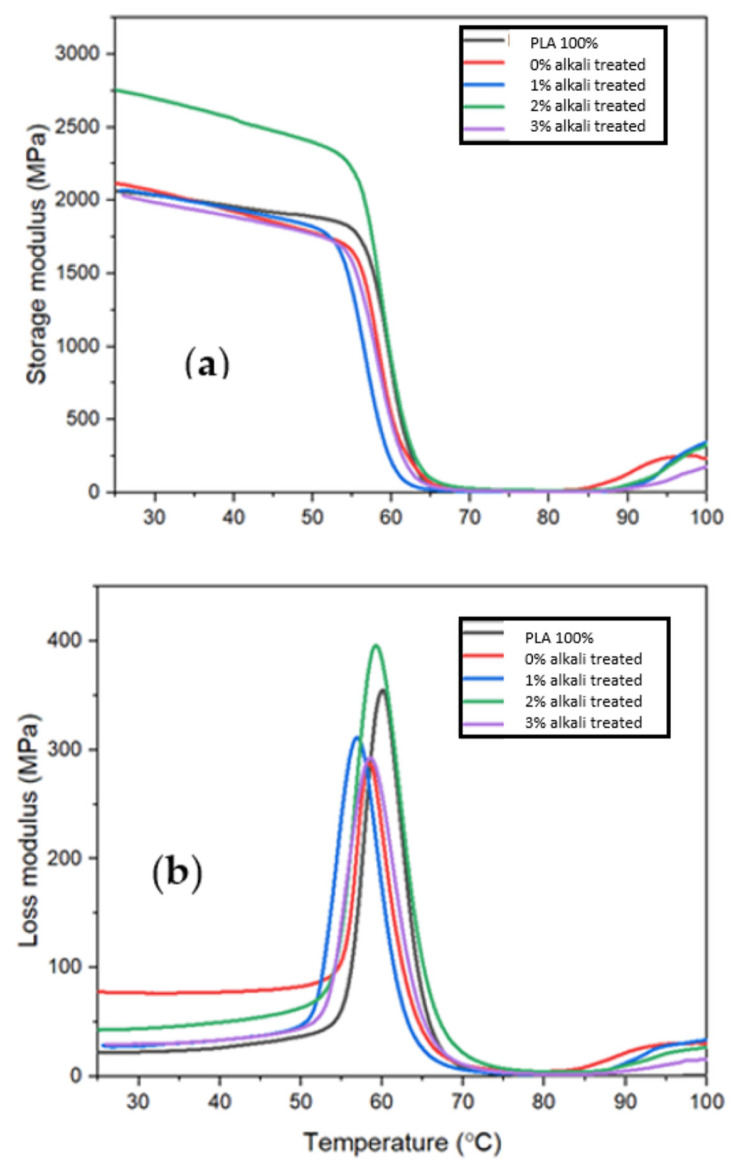
(**a**) Storage moduli of pure PLA, untreated, and alkali-treated miswak fiber/PLA composites; (**b**) Loss moduli of neat PLA, untreated, and treated miswak fiber/PLA composites.

**Figure 9 polymers-15-02228-f009:**
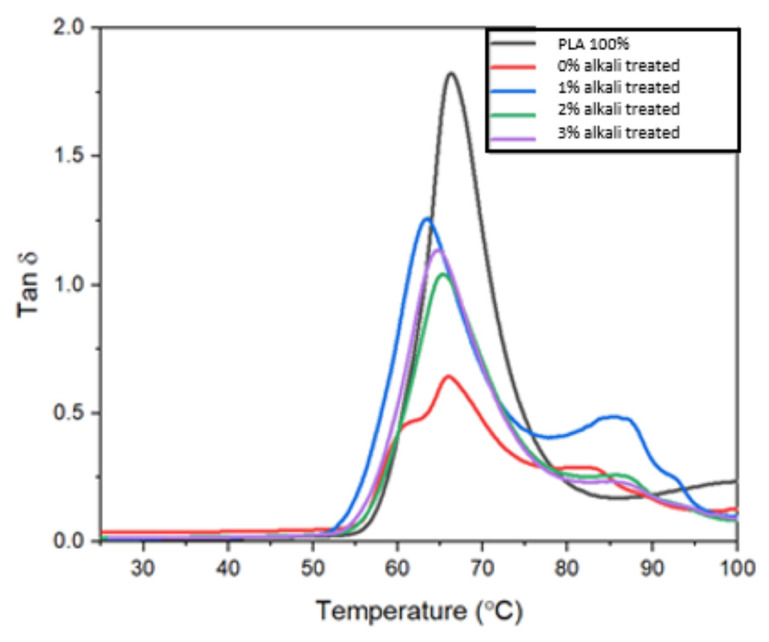
Tan δ of neat PLA, untreated, and alkali-treated miswak fiber/PLA composites.

**Table 1 polymers-15-02228-t001:** Properties of polylactic acid Ingeo 2003D.

No	Properties	Value
1	Density	1.24 g/cm^3^
2	MFR g/10 min (210 °C, 2.16 kg)	6 g/10 min
3	Clarity	Transparent
4	Melt Temperature	160 °C
5	Viscosity	4.0 Cp
6	Molecular Weight	62,000 g/mol
7	Moisture Content	≤0.025%
8	Glass Transition Temperature	55–60 °C

**Table 2 polymers-15-02228-t002:** Properties of miswak fiber.

No	Properties	Value
1	Density (g/cm^3^)	1.28
2	Cellulose (%)	31–70
3	Lignin (%)	7–26
4	Hemicellulose (%)	10–25
5	Moisture Absorption (%)	9.30

**Table 3 polymers-15-02228-t003:** Storage modulus, loss modulus, and damping value of untreated and alkali-treated MF/PLA composites.

Sample	PLA 100%	MF 30% + PLA 70%			
		Alkali-Treated			
		0%	1%	2%	3%
**Storage Modulus, E’ (GPa)**	1.8	1.5	1.8	2.4	1.7
**Loss Modulus, E” (Gpa)**	0.354	0.287	0.311	0.395	0.292
**Tan δ**	1.8	0.6	1.2	1	1.1
**T_g_ (°C)**	66	63	69	68	67

## Data Availability

The raw/processed data required to reproduce these findings cannot be shared at this time as the data also forms part of an ongoing study.

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
