# Peer review of "Effect of Alkaline Treatment on Mechanical and Thermal Properties of Miswak (Salvadora persica) Fiber-Reinforced Polylactic Acid"

_polymers, 2023, doi:10.3390/polym15092228_

Round 1
Reviewer 1 Report
The paper “Effect Of Alkaline Treatment On Mechanical and Thermal Properties of Miswak (Salvadora Persica) Fiber Reinforced Polylactic Acid” describes the preparation miswak fibres treated with various concentrations of sodium hydroxide (1wt%, 2wt%, and 3wt%) and the properties of some composites achieved by mixing the treated or untreated miswak fibres (30 wt%) and polylactic acid (70 wt%). By analyzing the presented results, it seems that the resulting composites do not display any improvement of the characteristics and the purpose of the study was not clearly identified to highlight the importance of this work, so the scientific importance of the study is low.
A part of my observations related to this manuscript are:
-First of all, in the Introduction, the authors should clearly state the novelty and the importance of the work, especially to highlight the advantages of the study.
- The modifications arose in the FTIR spectra by changing the composition are not very obvious, maybe some parts of the spectra should be enlarged and the peaks of interest highlighted. Also, the explanation of FTIR spectra is confusing. For example, the peak at 1180 cm-1 is attributed either to “the peak at 1180 cm-1 relates to the C-O stretching of aliphatic ether groups of hemicelluloses”, to “as do the peaks at 1180 cm-1, which are assigned to a C-O stretching vibration of acetyl group in lignin component” and “An increase in infrared transmittance at 1180 cm-1 following alkali treatment is clearly indicative of an increase in aromatic ether (C-O) and a reduction in hemicellulose and lignin”. If hemicellulose and lignin proportions are reduced, from which there is an increase of aromatic ether (C-O) sequences? Where in the structure of cellulose, aromatic ether (C-O) sequences are located? Perhaps a chemical representation of the alkaline treatment of miswak fibres would be more explicit.
- To better illustrate the modifications of the miswak fibres after alkali treatment, the FTIR spectra and SEM images of the fibres without PLA matrix should be provided. Also, the XRD investigation of the fibres should be added, to monitor the crystallinity modification after the alkaline hydrolysis.
- In general, the reinforcement of polymeric materials with various counterparts aims to improve the properties of the resulting materials, especially the mechanical ones. In the present study, for all the mechanical parameters investigated (tensile strength, tensile modulus, elongation at break), no improvement is observed with the introduction of miswak fibres, then what is the purpose for which these materials were made? Also, in the describing of the tensile testing experiments, the dimensions of the samples (120 x 120 x 2 mm) seems odd, please verify.
- In the SEM images, the scale bar is too small and the dimension is not visible.
-According to TGA investigation, the neat PLA sample displayed the highest thermal stability. The same question as for mechanical parameters, what is the purpose for which these materials were made?
- Figures 9 and 10 can be combined into a single figure (a, b). Same for Figures 11 and 12.
- In the figures where it is the case, the differences between the samples should be better highlighted.
In the manuscript, only some minor language corrections have to be done.
Author Response
Reviewer 1
1) The paper “Effect Of Alkaline Treatment On Mechanical and Thermal Properties of Miswak (Salvadora Persica) Fiber Reinforced Polylactic Acid” describes the preparation miswak fibres treated with various concentrations of sodium hydroxide (1wt%, 2wt%, and 3wt%) and the properties of some composites achieved by mixing the treated or untreated miswak fibres (30 wt%) and polylactic acid (70 wt%). By analyzing the presented results, it seems that the resulting composites do not display any improvement of the characteristics and the purpose of the study was not clearly identified to highlight the importance of this work, so the scientific importance of the study is low. A part of my observations related to this manuscript are:
To justify on this statement, result findings are still under research with further improvements will need to be made. The purpose for blending natural fiber with PLA is to reduce the cost of production. I had explained in detail on pricing in the introduction part. This research is targeted until end product to develop ‘Miswak Holder’ by using 3D printing machine. Due to this, the cost of raw materials is controlled.
The use of single biodegradable material such as polylactic acid incurred high cost compare with conventional polymer. Price of PLA (2.2 $/kg) are found to be higher than of fossil-based plastics (0.95–1.7 $/kg) and around two times more expensive than PP (1.2 $/kg). Therefore, compounding the biodegradable polymer with natural fibre is an alternative to control the cost of production. This research and developments continue, economies that takes accountability on the cost of raw material and also the processing.
2) First of all, in the Introduction, the authors should clearly state the novelty and the importance of the work, especially to highlight the advantages of the study.
I had added the paragraph on the introduction part:
The performance and properties of composite materials are determined by the characteristics of the individual components and their compatibility at the interface. Natural fibres has possess some drawbacks when blending with polymer which are high moisture content and poor compatibility. Brittle matrices with significant moisture absorption and swelling tend to develop cracks on the composite surface. To effectively improve interfacial interactions and provide desired characteristics, their surface properties must be modified adequately. Alkali treatment on fibre surface is one of alternative to overcome this issues. Alkali treatment can also bring different functional groups to the surface of natural fibres, and these functional groups can create strong covalent connections with the matrix to create a strong fiber/matrix contact. When just moderate mechanical properties are needed, natural fibre composites are a great alternative.
However, the use of single biodegradable material such as polylactic acid incurred high cost compare with conventional polymer. Price of PLA (2.2 $/kg) are found to be higher than of fossil-based plastics (0.95–1.7 $/kg) and around two times more expensive than PP (1.2 $/kg). Therefore, compound the biodegradable polymer with natural fibre is an alternative to control the cost of production. This research and developments continue, economies that takes accountability on the cost of raw material and also the processing.
The aim of this work was to explore the possibility of improving the effective of alkaline treatment of miswak fibre on the FTIR, tensile, TGA and DMA characteristics reinforced polylactic acid composites.
3) The modifications arose in the FTIR spectra by changing the composition are not very obvious, maybe some parts of the spectra should be enlarged and the peaks of interest highlighted. Also, the explanation of FTIR spectra is confusing. For example, the peak at 1180 cm-1 is attributed either to “the peak at 1180 cm-1 relates to the C-O stretching of aliphatic ether groups of hemicelluloses”, to “as do the peaks at 1180 cm-1, which are assigned to a C-O stretching vibration of acetyl group in lignin component” and “An increase in infrared transmittance at 1180 cm-1 following alkali treatment is clearly indicative of an increase in aromatic ether (C-O) and a reduction in hemicellulose and lignin”. If hemicellulose and lignin proportions are reduced, from which there is an increase of aromatic ether (C-O) sequences? Where in the structure of cellulose, aromatic ether (C-O) sequences are located? Perhaps a chemical representation of the alkaline treatment of miswak fibres would be more explicit.To better illustrate the modifications of the miswak fibres after alkali treatment, the FTIR spectra and SEM images of the fibres without PLA matrix should be provided. Also, the XRD investigation of the fibres should be added, to monitor the crystallinity modification after the alkaline hydrolysis.
I had rewritten the whole paragraph. XRD analysis can’t be conducted due to the machine still under maintenance. I will provide the data for next submission research paper.
4) In general, the reinforcement of polymeric materials with various counterparts aims to improve the properties of the resulting materials, especially the mechanical ones. In the present study, for all the mechanical parameters investigated (tensile strength, tensile modulus, elongation at break), no improvement is observed with the introduction of miswak fibres, then what is the purpose for which these materials were made? Also, in the describing of the tensile testing experiments, the dimensions of the samples (120 x 120 x 2 mm) seems odd, please verify.
I have included in the introduction part the purpose of this study. The application of this study is to develop a miswak holder. The use of a single biodegradable polymer may cause high costs for raw materials. In order to control the price of raw materials as well as production cost, natural fibre was added into the polymer. Even though the properties are not increased much, I believe the purpose of this study was achieved.
I revised the measurement of tensile specimen to 150 x 23 x 2 mm in length, width, and thickness.
5) In the SEM images, the scale bar is too small and the dimension is not visible.
I added the scale bar on the image
6) According to TGA investigation, the neat PLA sample displayed the highest thermal stability. The same question as for mechanical parameters, what is the purpose for which these materials were made?
I have included in the introduction part the purpose of this study. The application of this study is to develop a miswak holder. The use of a single biodegradable polymer may cause high costs for raw materials. In order to control the price of raw materials as well as production cost, natural fibre was added into the polymer. Even though the properties are not increasing much, I believe the purpose of this study was achieved.
7) Figures 9 and 10 can be combined into a single figure (a, b). Same for Figures 11 and 12.
I merged both Figures as suggested

Reviewer 2 Report
The research work is focused on the thermal and mechanical properties of PLA matrix composites and reinforced with short miswak fibers. The authors highlight the effect of an alkaline pretreatment of fibers on the ultimate performance of composite products.
While appreciating the sustainability aspects of the research, major revisions are compulsory to deepen some points and clean up the text from typewriting errors still present in it.
In particular, it is suggested to review the following points.
Lines 34-36: Aside from the misspelled "improved" instead of "improve", while the meaning of the sentence starting with “To effectively …” is easily guessed by readers familiar with this topic, it is recommended that authors review it. This assertion, in the light of the previous sentence, seems to indicate the need to modify the surface of composite materials. Actually, there is a need to modify the surface properties of the main constituent phases (matrix and reinforcement) to improve the quality of the interface but this concept is not clearly explained.
Line 100: erroneously it is written "pallets" instead of "pellets". Please correct.
Paragraph 2.3: since this is a research article and not a technical report, it is advisable to include details regarding the equipment (mixer, press, etc.) used for the production of the samples.
Line 117: please specify details about the equipment used to produce dog-bone specimens.
Line 166: part of the verb is probably missing. Specifically, perhaps the sentence should end with "are eliminated". Please check.
Paragraph 3.5 - Lines 297-298: This sentence is superfluous also in the light of the introductory sentence of this paragraph.
Line 305: Figure 10 is erroneously mentioned but this element refers to the comparison of DTG curves.
In general, the manuscript does not report DMA graphs relating to composites containing untreated fibers which instead must be included in the manuscript. In other words, it is recommended to supplement the text with comparison graphs that support the discussion of the results.
Line 309: it is suggested to replace the second word "fibre" with the word "one" to avoid repetitions.
Line 311: the text reports that the alkaline treatment is "…the most efficient method…" but this cannot be said also because the research only considered this surface treatment. It is suggested to replace that section of the sentence with "…an effective method…".
Line 319: it is worth specifying that the graph of Figure 12 includes the curves of composites with pre-treated miswak fibers.
Line 322: Text talks about "...PLA matrix's crystalline areas...". On what basis are structural aspects of the matrix cited? Why do the authors expect that the pretreatment of the reinforcing fibers can also influence the crystallinity of the matrix? Are there experimental evidences already available and not included in this manuscript? Please explain.
Lines 335-336: It is written "...Figure 13 demonstrates that the damping peak in untreated composites declines." However, Figure 13 does not report the loss factor curves for composites containing untreated fibers. Please integrate the Figure to support the discussion.
The language is fine but the text requires careful proofreading to eliminate typos and repetitions.
Author Response
Reviewer 2
The research work is focused on the thermal and mechanical properties of PLA matrix composites and reinforced with short miswak fibers. The authors highlight the effect of an alkaline pretreatment of fibers on the ultimate performance of composite products. While appreciating the sustainability aspects of the research, major revisions are compulsory to deepen some points and clean up the text from typewriting errors still present in it.
In particular, it is suggested to review the following points.
1) Lines 34-36: Aside from the misspelled "improved" instead of "improve", while the meaning of the sentence starting with “To effectively …” is easily guessed by readers familiar with this topic, it is recommended that authors review it. This assertion, in the light of the previous sentence, seems to indicate the need to modify the surface of composite materials. Actually, there is a need to modify the surface properties of the main constituent phases (matrix and reinforcement) to improve the quality of the interface but this concept is not clearly explained.
Sentence Before: The performance and properties of composite materials are determined by the characteristics of the individual components and their compatibility at the interface. To effectively improved interfacial interactions and provide desired characteristics, their surface properties must be modified adequately. When just moderate mechanical properties are needed, natural fibre composites are a great alternative.
Sentence After: The performance and properties of composite materials are determined by the characteristics of the individual components and their compatibility at the interface. Natural fibres has possess some drawbacks when blending with polymer which are high moisture content, poor compatibility. Brittle matrices with significant moisture absorption and swelling tend to develop cracks on the composite surface. To effectively improve interfacial interactions and provide desired characteristics, their surface properties must be modified adequately. Alkali treatment on fibre surface is one of alternative to overcome this issues. Alkali treatment can also bring different functional groups to the surface of natural fibres, and these functional groups can create strong covalent connections with the matrix to create a strong fiber/matrix contact. When just moderate mechanical properties are needed, natural fibre composites are a great alternative
2) Line 100: erroneously it is written "pallets" instead of "pellets". Please correct.
Had made corrections to the spelling
3) Paragraph 2.3: since this is a research article and not a technical report, it is advisable to include details regarding the equipment (mixer, press, etc.) used for the production of the samples.
I Had written the brand of the equipment in the sentence except for the pulverizer because I don't have info about the machine.
The PLA pallets are initially desiccated in a furnace (ESCO ISOTHERM model range temperature 0-150°C) at 80 °C to prevent excessive hydrolysis, which could compromise the polymer's physical properties. Then, a counter rotating internal mixer (BRABENDER) was used to combine PLA with miswak fibre at 160 °C and 60 RPM for 15 minutes, or until a homogeneous compound was obtained. Using a pulverize machine, the compound was reduced to small fragments prior to being compressed. The composite was then compressed into a testing board using molded under hot press (TECHNOPRESS) for 4 minutes at 160 °C, followed by 3 minutes of cooling period. The specimens were then shaped in accordance with the desired characterization tests.
4) Line 117: please specify details about the equipment used to produce dog-bone specimens.
Sentence Before: The composite was tested with tensile testing using a 5kN Bluehill INSTRON Universal Testing Machine. The test was carried out according to ASTM standard D-638. With dimensions of 120 x 120 x 2 mm in length, width, and thickness the specimen was molded into a dog bone form using a plastic moulder machine.
The crosshead speed was set at 2.0mm/min, and the composites were held at a gauge length of 30mm. All specimen were stored in the conditioning room, and the test was conducted at 22 °C with a relative humidity (RH) of 55%. For each test condition, seven specimens were examined.
Revise sentence: The composite was tested with tensile testing using a 5kN Bluehill INSTRON Uni-versal Testing Machine. The test was carried out according to ASTM standard D-638. With dimensions of 120 x 120 x 2 mm in length, width, and thickness the specimen was molded into a dog bone form using a MT1130 die-cutter (MinEuro, Turin, Italy) equipped with a metallic dog-bone shaped cutting edge. The sample was gripped using the metallic dog bone shape before push the sample through the cutter.
The crosshead speed was set at 2.0mm/min, and the composites were held at a gauge length of 30mm. All specimen were stored in the conditioning room, and the test was conducted at 22 °C with a relative humidity (RH) of 55%. For each test condition, seven specimens were examined.
5) Line 166: part of the verb is probably missing. Specifically, perhaps the sentence should end with "are eliminated". Please check.
I had made correction on the verb
6) Paragraph 3.5 - Lines 297-298: This sentence is superfluous also in the light of the introductory sentence of this paragraph.
I added on the introduction of DMA testing in the first 3.5 paragraphs.
7) Line 305: Figure 10 is erroneously mentioned but this element refers to the comparison of DTG curves.
The figure was changed to 7b and discussed DTG curves
8) In general, the manuscript does not report DMA graphs relating to composites containing untreated fibers which instead must be included in the manuscript. In other words, it is recommended to supplement the text with comparison graphs that support the discussion of the results.
I had included Table 3 and an explanation of both treated and untreated fibre
9) Line 309: it is suggested to replace the second word "fibre" with the word "one" to avoid repetitions.
I replaced the word
10) Line 311: the text reports that the alkaline treatment is "…the most efficient method…" but this cannot be said also because the research only considered this surface treatment. It is suggested to replace that section of the sentence with "…an effective method…".
I replaced the word as suggested
11) Line 319: it is worth specifying that the graph of Figure 12 includes the curves of composites with pre-treated miswak fibers.
The graph only shows the difference between untreated and treated composite. 0% refers to untreated fibre.
12) Line 322: Text talks about "...PLA matrix's crystalline areas...". On what basis are structural aspects of the matrix cited? Why do the authors expect that the pretreatment of the reinforcing fibers can also influence the crystallinity of the matrix? Are there experimental evidences already available and not included in this manuscript? Please explain.
I revised the sentence. The crystallinity is referred to fibre, not PLA.
As reported by Siakeng et al. (2018), surface treatment partially dissolved lignin and hemicellulose contained in the fiber, thus increasing the amount of exposed cellulose, and owing to this, the crystallinity of MF was modified with alkali treatment. Increasing crystallinity, the modulus of PLA also increases The correlation between an increase in Tg and a reduction in the mobility of the matrix chains is indicative of increased interfacial adhesion between the fibre and matrix.

Reviewer 3 Report
In this paper, the authors have investigated the effect of alkaline treatment on the mechanical and thermal properties of miswak fiber-reinforced polylactic acid. The authors proposed insight into plant-based fiber in the polylactic acid matrix to enhance the physical properties of the polymers. However, the discussions for this version to make the achievement of these fields are weak. Therefore, the authors need the revision the manuscript for publication in Polymers. Some questions and suggestions are as followed;
[1] The Introduction section is too weak. We suggest that authors should add more background information and related literatures in the Introduction section.
[2] We suggest that authors should add the scale bar in SEM Figures, as possible.
[3] We suggest that authors should merge Figures 3 and 4 into one Figure composed of Figures (a) and (b). We suggest that authors should merge Figures 7 and 8 into one Figure composed of Figures (a) and (b). We suggest that authors should merge Figures 9 and 10 into one Figure composed of Figures (a) and (b). We suggest that authors should merge Figures 11 and 12 into one Figure composed of Figures (a) and (b).
[4] The resolution of some Figures is too low. Therefore, we suggest that the authors should change the Figures with higher image quality to increase the understanding of this manuscript by journal readers.
[5] We judged that authors should cite the related literature including recent studies in the References section.
[6] The form of references described in the References part does not match the guideline of the “Polymers” journal. The authors should revise the references’ form accurately.
[7] We suggest that authors should improve English expression in the whole manuscript as much as possible. We suggest that authors should correct the typographic error such as superscript/subscript and case-sensitive and unit and spacing word issues in the whole manuscript. For example, authors should change the title and subtitle of the manuscript under case-sensitive correction in the revised manuscript.
We suggest that authors should improve English expression in the whole manuscript as much as possible. We suggest that authors should correct the typographic error such as superscript/subscript and case-sensitive and unit and spacing word issues in the whole manuscript. For example, authors should change the title and subtitle of the manuscript under case-sensitive correction in the revised manuscript.
Author Response
Reviewer 3
[1] The Introduction section is too weak. We suggest that authors should add more background information and related literatures in the Introduction section.
I had added some literature study on the purpose of blending PLA and natural fibre in the introduction part.
[2] We suggest that authors should add the scale bar in SEM Figures, as possible.
An adjustment has been made by adding a scale bar in the SEM figure
[3] We suggest that authors should merge Figures 3 and 4 into one Figure composed of Figures (a)and(b)
Figures have been merged
[4] We suggest that authors should merge Figures 7 and 8 into one Figure composed of Figures (a) and (b)
Figures have been merged
[5] We suggest that authors should merge Figures 9 and 10 into one Figure composed of Figures (a) and (b)
Figures have been merged
[6] Figures have been merged (b). We suggest that authors should merge Figures 11 and 12 into one Figure composed of Figures (a) and (b)
Figure have been merged
[4] The resolution of some Figures is too low. Therefore, we suggest that the authors should change the Figures with higher image quality to increase the understanding of this manuscript by journal readers.
Had a change with high-resolution Figures
[5] We judged that authors should cite the related literature including recent studies in the References section.
I had added few more recent study related to this topic.
[6] The form of references described in the References part does not match the guideline of the “Polymers” journal. The authors should revise the references’ form accurately.
Had made revisions to the references format
[7] We suggest that authors should improve English expression in the whole manuscript as much as possible. We suggest that authors should correct the typographic error such as superscript/subscript and case-sensitive and unit and spacing word issues in the whole manuscript. For example, authors should change the title and subtitle of the manuscript under case-sensitive correction in the revised manuscript.
I had revised the manuscript as suggested.

Round 2
Reviewer 1 Report
The authors have answered most of the problems indicated.
In the manuscript there are still some grammatical or sentence organization mistakes.
Reviewer 2 Report
The revised version of the manuscript is acceptable.
The language is fine.
Reviewer 3 Report
The current version is acceptable for publication in Polymers.
The current version is acceptable for publication in Polymers.